# Protective effect of Mediterranean-type glucose-6-phosphate dehydrogenase deficiency against *Plasmodium vivax* malaria

Ghulam R Awab[1,2], Fahima Aaram[3], Natsuda Jamornthanyawat[4], Kanokon Suwannasin[4], Watcharee Pagornrat[4], James A Watson[1,5], Charles J Woodrow[1,5], Arjen M Dondorp[1,5], Nicholas PJ Day[1,5], Mallika Imwong[4], Nicholas J White[1,5]*

[1]Mahidol Oxford Tropical Medicine Research Unit (MORU), Faculty of Tropical Medicine, Mahidol University, Bangkok, Thailand; [2]Nangarhar Medical Faculty, Jalalabad, Afghanistan; [3]Kabul Medical University, Kabul, Afghanistan; [4]Department of Molecular Tropical Medicine and Genetics, Faculty of Tropical Medicine, Mahidol University, Bangkok, Thailand; [5]Centre for Tropical Medicine and Global Health, Nuffield Department of Medicine, University of Oxford, Oxford, United Kingdom

**Abstract** X-linked glucose-6-phosphate dehydrogenase (G6PD) deficiency is the most common human enzymopathy. The severe Mediterranean variant (G6PD Med) found across Europe and Asia is thought to confer protection against malaria, but its effect is unclear. We fitted a Bayesian statistical model to observed G6PD Med allele frequencies in 999 Pashtun patients presenting with acute *Plasmodium vivax* malaria and 1408 population controls. G6PD Med was associated with reductions in symptomatic *P. vivax* malaria incidence of 76% (95% credible interval [CI], 58–88) in hemizygous males and homozygous females combined and 55% (95% CI, 38–68) in heterozygous females. Unless there is very large population stratification within the Pashtun (confounding these results), the G6PD Med genotype confers a very large and gene-dose proportional protective effect against acute vivax malaria. The proportion of patients with vivax malaria at risk of haemolysis following 8-aminoquinoline radical cure is substantially overestimated by studies measuring G6PD deficiency prevalence in healthy subjects.

*For correspondence:
nickw@tropmedres.ac

Competing interests: The authors declare that no competing interests exist.

## Introduction

In red blood cells, glucose-6-phosphate dehydrogenase (G6PD; EC 1.1.1.49) is the only source of reduced nicotinamide adenine dinucleotide phosphate (NADPH) (*Beutler, 1994*; *Luzzatto and Arese, 2018*). G6PD deficiency reflects instability, not absence, of this enzyme, which is essential for normal cellular function. Mammalian red blood cells lack nuclei and the necessary protein synthetic pathway, and so, unlike nucleated cells, they cannot replenish degraded G6PD. In G6PD deficiency, the active enzyme content of erythrocytes declines markedly as they age. NADPH is essential for the maintenance of oxidant defences. Thus, as G6PD deficient red cells age, they become increasingly susceptible to oxidant haemolysis. G6PD deficiency is the most common enzyme abnormality in humans. It is found across the malaria-endemic world with mutant gene prevalences up to 35% (average 8–10%) (*Howes et al., 2012*). There are over 200 different polymorphic variants, most of which result in enzyme deficiency, but the degree of deficiency (from accelerated enzyme degradation) and thus vulnerability to oxidant haemolysis varies substantially among the different genotypes. G6PD deficiency is X-linked, so males are either normal or fully deficient. Women display these two

phenotypes (normal or homozygous deficiency) as well as intermediate deficiency (heterozygotes). The heterozygote females are genetic mosaics as a result of early embryonic random X-chromosome inactivation (Lyonisation). Their blood contains a mixture of G6PD-normal and G6PD-deficient erythrocytes. Overall, at a population level, the proportion averages 50:50 of each cell type, but there is inter-individual variation, so in some heterozygotes, the large majority of erythrocytes are G6PD deficient. The high prevalences of G6PD deficiency in tropical areas, particularly in Africa, and in areas where malaria was once endemic, suggest that G6PD deficiency confers protection either against malaria or its adverse effects. But the mechanism of protection and its extent is unclear. This has been a subject of controversy and divergent opinion, with no clear conclusion. Claims have been made that there is no malaria protective effect provided by G6PD deficiency or that protective effects are seen in female heterozygotes only, or in male hemizygotes only, or in both (*Mbanefo et al., 2017*; *Bienzle et al., 1972*; *Ruwende et al., 1995*; *Guindo et al., 2007*; *Mombo et al., 2003*; *Lopera-Mesa et al., 2015*; *Uyoga et al., 2015*; *Clarke et al., 2017*). Most of the studies addressing this question have focussed on falciparum malaria in Africa where the majority of evidence supports a protective effect against severe malaria, particularly in female heterozygotes (*Uyoga et al., 2015*; *Clarke et al., 2017*). Whether male hemizygotes and female homozygotes are protected is unclear with evidence both for and against. A recent meta-analysis of 28 studies suggested a moderate protective effect against uncomplicated falciparum malaria (odds ratio [OR]: 0.77; 95% credible interval [CI], 0.59–1.02), but this estimate could be affected by publication biases (*Mbanefo et al., 2017*).

In 2002, Richard Carter and Kamini Mendis suggested that the evolutionary force selecting G6PD deficiency could have been either *Plasmodium falciparum* or *Plasmodium vivax* (*Carter and Mendis, 2002*). Historically, *P. vivax* had a wider geographic distribution, although it has now been eradicated from North America, Europe, and Russia. Elsewhere in the Americas, the horn of Africa, Asia, and Oceania, *P. vivax* has become the predominant cause of malaria in recent years. In general, the variants of G6PD deficiency that are prevalent in these areas where *P. vivax* infections occur, or once occurred, are more severe than the common ('A−') variant prevalent in the sub-Saharan African populations (in whom *P. vivax* malaria is rare and *P. falciparum* comprises the large majority of malaria infections), and in people with their genetic origin there. The most severe of the commonly found G6PD variants is the 'Mediterranean' variant ('G6PD Med'). This results from a single C-T transition at nucleotide position 563, causing a serine phenylalanine replacement at amino acid position 188. G6PD Med is the predominant genotype in the Pashtun who live in Afghanistan, Pakistan, and India (*Bouma et al., 1995*; *Leslie et al., 2010*; *Jamornthanyawat et al., 2014*). An earlier retrospective study conducted in Afghan refugees in North-Western Pakistan (*Leslie et al., 2010*) suggested that G6PD Med hemizygotes were protected against vivax malaria, but there were too few observations to substantiate a trend to lower infection rates in female heterozygotes. To characterise the possible protective effects of G6PD Med against *P. vivax* malaria, we conducted a retrospective analysis of case–control data from clinical studies on vivax malaria, and epidemiological studies of G6PD deficiency that we have conducted in Afghanistan over the past 10 years. These data were then combined in a meta-analysis using all previously published data on G6PD deficiency in people of Pashtun ethnicity living in malaria-endemic areas.

## Results

### Retrospective case–control study

In total, 764 Pashtun patients presenting with acute vivax malaria (304 males, 460 females) and 699 Pashtun controls (342 males, 357 females) were studied; 236 healthy males came from the epidemiology study reported previously (*Awab et al., 2017*) and the remaining control subjects came from the same locations as the clinical malaria studies (*Table 1*; *Figure 1*). In the controls, the allele frequency of G6PD Med was estimated to be 7.8% (95% credible interval [CI], 6.3–9.5) under the assumption of Hardy–Weinberg equilibrium. There was no evidence of departure from Hardy–Weinberg in the controls (p=0.9). The proportions of G6PD Med male hemizygotes and female heterozygotes were substantially lower in patients with acute vivax malaria than in people incidentally visiting clinics or vaccination centres who did not have malaria (*Table 1*). Only 1.6% (5 of 304) of males with vivax malaria were hemizygotes (risk ratio [RR]; 95% CI, 0.12 [0.08–0.51]), while in the females with

**Table 1.** Summary of all case–control data included in the meta-analysis.

*Data from this report; 23 of 236 male controls from the earlier epidemiological study (*Jamornthanyawat et al., 2014*) and 5 of 106 male controls from the later studies were hemizygotes.

|  |  | General population (controls) | | | | P. vivax malaria (cases) | | | |
|---|---|---|---|---|---|---|---|---|---|
|  |  | Awab et al* | *Leslie et al., 2010* | *Bouma et al., 1995* | Total | Awab et al* | *Leslie et al., 2010* | *Bouma et al., 1995* | Total |
| Males | Hemizygous | 28 | 31 | 25 | 84 | 5 | 2 | 0 | 7 |
|  | Normal | 314 | 285 | 214 | 813 | 299 | 155 | 0 | 454 |
| Females | Homozygous | 2 | 2 | 0 | 4 | 3 | 0 | 0 | 3 |
|  | Heterozygous | 50 | 26 | 0 | 76 | 32 | 6 | 0 | 38 |
|  | Normal | 305 | 126 | 0 | 431 | 425 | 72 | 0 | 497 |

vivax malaria, 7.0% (32 of 460) (RR 0.61 [0.39–0.95]) and 0.7% (3 of 460) were heterozygotes and homozygotes, respectively. Under the Bayesian model, assuming Hardy–Weinberg equilibrium,

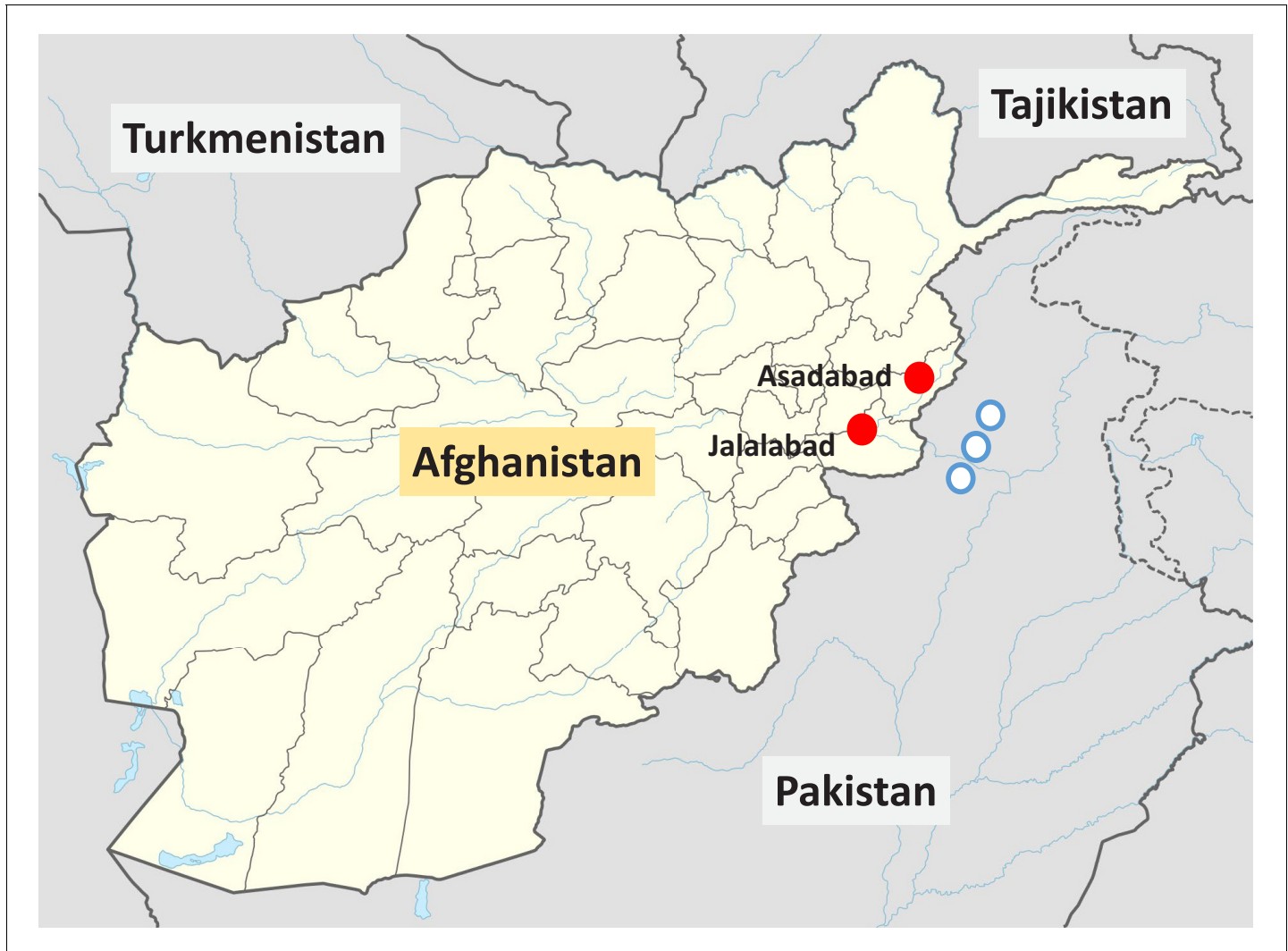

**Figure 1.** Locations of the two vivax malaria clinical study sites in Eastern Afghanistan from the present study (red circles), and the approximate locations of the villages in the North-West frontier province of Pakistan where Afghan Pashtun refugees were enrolled in vivax malaria clinical trials and later included in case–control studies (*Leslie et al., 2010*) (blue circles).

these results suggest that G6PDd Med hemizygous males and homozygous females have 68% protection (95% CI, 39–85; i.e. relative reduction, given by 1-α in the model) against acute *P. vivax* malaria, and G6PDd Med heterozygous females have 51% protection (95% CI, 28–67, given by 1-β in the model).

Geometric mean (range) *P. vivax* parasite densities were very similar in patients with wild-type G6PD (2099; 80–67,000 parasites/μL; N = 570) and female G6PD Med heterozygotes (2064; 310–26,110 parasites/μL; N = 25) and were slightly lower in the six patients with quantitative parasite counts who were G6PD Med hemizygotes or homozygotes (922; 150–3720 parasites/μL; p=0.14 from a logistic regression).

## Effect of a haemoglobin exclusion criterion

In our earlier studies, patients with moderate or severe anaemia (haemoglobin <8 g/dL) were excluded, which could have biased our results. We therefore compared distributions of haemoglobin concentrations in the earlier clinical trials, in which this exclusion was applied, with those in the more recent studies in which there was no anaemia exclusion criterion (Figure 3). This indicated that 9.6% (29 of 302) patients would have been excluded if the earlier study threshold had been applied. This is predicted to have resulted in exclusion of a single female heterozygote. There was no significant difference in the haemoglobin concentrations at presentation comparing G6PD Med hemizygotes or homozygotes, heterozygotes, or wild type (Figure 3). This shows that exclusion of anaemic patients in the first part of the study did not affect the interpretation of the protective effect of G6PD deficiency on vivax malaria.

## Meta-analysis

These data were then combined with data from the two previously reported studies on the prevalence of G6PD deficiency in the Pashtun ethnic group from Afghanistan. One was in healthy subjects only (*Bouma et al., 1995*), and the other included both vivax malaria patients and matched healthy control subjects (*Leslie et al., 2010*; *Table 1*). The meta-analysis of all three studies gave a slightly higher G6PD Med allele frequency of 8.8% (95% CI, 7.6–10.1). The overall protective effect in male hemizygotes and female homozygotes was estimated as 76% (95% CI, 58–88) and in female heterozygotes was 55% (95% CI, 38–68). The posterior distributions for these estimates from the meta-analysis are shown in *Figure 2*. The posterior probability was 0.98 that the protective effect observed in female heterozygotes was less than the protective effect observed in male hemizygotes and female homozygotes , suggesting that the protective effect is proportional to the gene dose.

## Discussion

Mediterranean-type glucose-6-phosphate dehydrogenase deficiency (G6PD Med) prevalent in the Pashtun provided a strong and gene-dose-related protective effect against *Plasmodium vivax* malaria. This is a much greater protective effect than observed against *P. falciparum* malaria elsewhere (*Mbanefo et al., 2017*; *Uyoga et al., 2015*). It is probably explained by two factors. First, the degree of enzyme deficiency with G6PD Med is substantially greater than in the common African A− variant, which has been the main genotype studied previously in falciparum malaria studies. Second, *P. vivax* is generally more sensitive to oxidant effects than *P. falciparum*. Compared with *P. falciparum*, asexual stages of *P. vivax* are more sensitive to oxidant drugs (i.e. artemisinins and synthetic peroxides and 8-aminoquinolines) (*Phyo et al., 2016*; *Pukrittayakamee et al., 2000*). *P. vivax* may therefore be more sensitive to the oxidant stresses associated with G6PD deficiency. This large study from Afghanistan confirms earlier findings from a case–control study in Afghan Pashtun refugees, based mainly on phenotyping. This earlier study showed clear evidence of protection against vivax malaria in male hemizygotes, but in the smaller subgroup of genotyped female heterozygotes the uncertainty around the estimated effect was large (adjusted odds ratio [AOR]: 0.4, 95% CI, 0.16–1.02) (*Leslie et al., 2010*). These earlier studies, and a smaller series from Iran with phenotyping (*Ebrahimipour et al., 2014*), are all consistent with the present series. Combined together they show clearly that G6PD Med provides a substantial gene dose proportionate protection against vivax malaria. The inference of the gene-dose effect in females assumes that hemizygote males and homozygote females are phenotypically identical, and, as expected from the Hardy–Weinberg equilibrium, there were few female homozygotes, so the majority of the protective signal in this

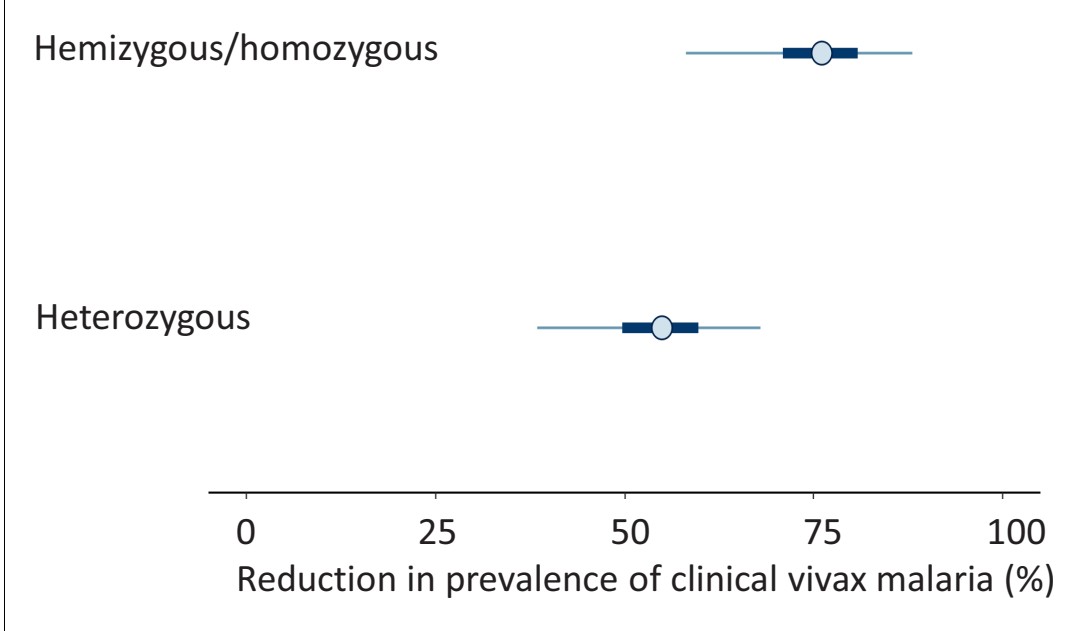

**Figure 2.** Results from the meta-analysis assessing the protective effect of the Mediterranean variant of G6PD deficiency against *Plasmodium vivax* malaria. The posterior distributions of 1-α (top: hemi/homozygotes) and 1-β (bottom: heterozygotes) are shown as percentages. These values represent the reduction in prevalence of clinical vivax malaria relative to G6PD normal individuals. The circles show the median estimates, with the 50% credible intervals shown by the thick blue lines and the 95% credible intervals shown by the thin blue lines.

combined group is from the hemizygote males. In a survey conducted in periurban Manaus, in the Amazon region of Western Brazil, where *P. vivax* is now the predominant (90%) cause of malaria, there was also a very strong protective effect of G6PD Med against self-reported previous malaria (AOR: 0.010) (*Santana et al., 2013*). In comparison the protective effect of G6PD A– in the same location was an order of magnitude weaker (AOR: 0.119). This marked protective benefit of G6PD deficiency against *P. vivax* infections is critically important for the assessment of population haemolytic risk associated with giving 8-aminoquinoline antimalarials for the radical cure for vivax malaria. These are the only drugs providing radical cure of vivax malaria (prevention of relapse), but they are underused because G6PD deficiency testing is usually not available and prescribers are naturally concerned about precipitating dangerous haemolysis. The proportion of patients with vivax malaria at risk of serious haemolysis with G6PD Med in these studies is nearly four times lower than would be predicted from gene frequencies in the healthy population.

G6PD Med is among the most severe of the polymorphic genetic G6PD variants. It is found across the malaria-endemic regions of the world, having evolved independently on several occasions (*Jamornthanyawat et al., 2014*; *Kurdi-Haidar et al., 1990*). There is also evidence for protection against vivax malaria from the moderate severity G6PD Mahidol variant. A study from western Thailand found that *P. vivax* densities were lower in hemizygote males and also in heterozygote females presenting to clinics with vivax malaria, but there was no corresponding effect in falciparum malaria, and there was no apparent protective effect against malaria illness (*Louicharoen et al., 2009*). However, a case–control study from Northern Myanmar in a population with a high prevalence (25.2%) of the G6PD Mahidol variant gave adjusted odds ratios for having acute vivax malaria of 0.213 (95% CI, 0.093–0.487) for male 487A hemizygotes, and 0.248 (0.110–0.561) for female heterozygotes (*Yi et al., 2019*). In a multi-site survey of G6PD deficiency in malaria patients in Cambodia, where G6PD Viangchan predominates (WHO class two but quantitatively similar to G6PD Mahidol), phenotypic severe deficiency (i.e. <10% of population normal) provided stronger protection against *P. vivax* than *P. falciparum* infections (OR: O.45; 95% CI, 0.32–0.64) (*Khim et al., 2013*). Conversely, Dewasurendra et al. in Sri Lanka reported evidence of hemizygote protection from malaria and lower parasite densities for *P. falciparum*, but not *P. vivax* (*Dewasurendra et al., 2015*).

A recent meta-analysis of 28 studies (*Mbanefo et al., 2017*) addressing the question of whether G6PD deficiency protects against malaria showed a moderate protective effect against uncomplicated falciparum malaria (OR: 0.77; 95% CI, 0.59–1.02). The degree of protection was similar in female heterozygotes (OR: 0.7; 95% CI, 0.57–0.87) and in male hemizygotes and female homozygotes (OR: 0.7; 95% CI, 0.46–1.07) but did not reach statistical significance in the latter group. There was evidence for publication bias towards significant findings in the uncomplicated malaria comparisons. The same meta-analysis also concluded that there was no statistically significant protective effect in severe malaria or in *P. vivax* malaria, although there were limited data to assess the protective effects in *P. vivax* infections.

The degree of protection conferred by G6PD Med against *P. vivax* illness estimated in this study is large; it is similar in magnitude to the well-described protection conferred by Hb AS (sickle cell heterozygotes) against falciparum malaria (*Williams et al., 2005*).

It is possible that G6PD deficiency confers no significant protection against uncomplicated *P. falciparum* malaria, but protects only against life-threatening illness. This may reflect either inhibition of parasite multiplication or a different protective mechanism. One consistent clinical feature of G6PD deficiency is an increased risk of anaemia in acute infections as the deficient red cell haemolyse (*Uyoga et al., 2015*; *Clarke et al., 2017*). As severe anaemia (Hb < 5 g/dL) is one of the criteria for defining severe malaria, this results in a higher proportion of patients with G6PD deficiency presenting with severe malarial anaemia, and therefore being diagnosed as having severe malaria. This can bias genetic association studies (*Watson et al., 2019*). It has been suggested that G6PD deficiency may protect specifically against cerebral malaria, but a simpler explanation is that, in the context of severe falciparum malaria, a rapid onset of moderate anaemia protects against life-threatening complications such as cerebral malaria (*White, 2018*; *Leopold et al., 2019*). In most areas, *P. falciparum* and *P. vivax* coexist. The interaction between the malaria species is complex. *P. falciparum* was once prevalent in malaria-endemic regions of Afghanistan, but it has now been all but eliminated. There were no coinfections identified in this study, so *P. falciparum* was not a confounder.

There are several limitations to this study. It was not designed prospectively as a case–control study. It combines results from a prospective epidemiology study conducted 6 years ago and prospective clinical trials and sampling of controls from the same centres from 2018. For security reasons, careful matching of cases and controls (other than for location) was not possible. We do not have additional genetic marker data that could be used to adjust for population stratification. However, in the earlier study by *Leslie et al., 2010*, conducted in refugee camps in Pakistan, careful matching was done, particularly with reference to tribe (within the Pashtun group) and location, and that study's findings are consistent with the present investigation. Nevertheless, it remains possible that uncharacterised variations between the different investigations and genetic heterogeneity within the Pashtun group are confounders. As many of the controls had non-malaria febrile illnesses, it is possible, although unlikely, that the prevalence of G6PD deficiency is higher in such patients to that in the general population. In the initial clinical studies reported here, and in the earlier studies in Pashtun refugees (*Leslie et al., 2010*), severe anaemia was an exclusion criterion. This could have reduced the proportion of G6PD-deficient patients, but there was no evidence for differences in presenting haemoglobin concentrations between G6PD-deficient and G6PD-normal patients (*Figure 3*). This study was confined to the G6PD Mediterranean genotype, so other G6PD polymorphisms were not studied, but these are unusual in the Pashtun. If there were undetected G6PD-deficient patients, then they would have diluted the apparent protective benefit. Potential confounders become progressively more relevant as effects become smaller.

Overall, this study shows that the G6PD Mediterranean genotype confers a very large and gene-dose proportional protective effect against vivax malaria. If G6PD Med is approximately at an equilibrium allele frequency in the Pashtun population (i.e. is under balancing selection), this implies that it must be harmful, i.e. malaria alone cannot create balancing selection. The main risks associated with G6PD deficiency are neonatal hyperbilirubinaemia and haemolysis following oxidant food and drugs, while the benefit is protection from malaria. Importantly for malaria treatment, the population risks associated with 8-aminoquinoline radical cure are greatly overestimated from G6PD prevalences in the healthy population.

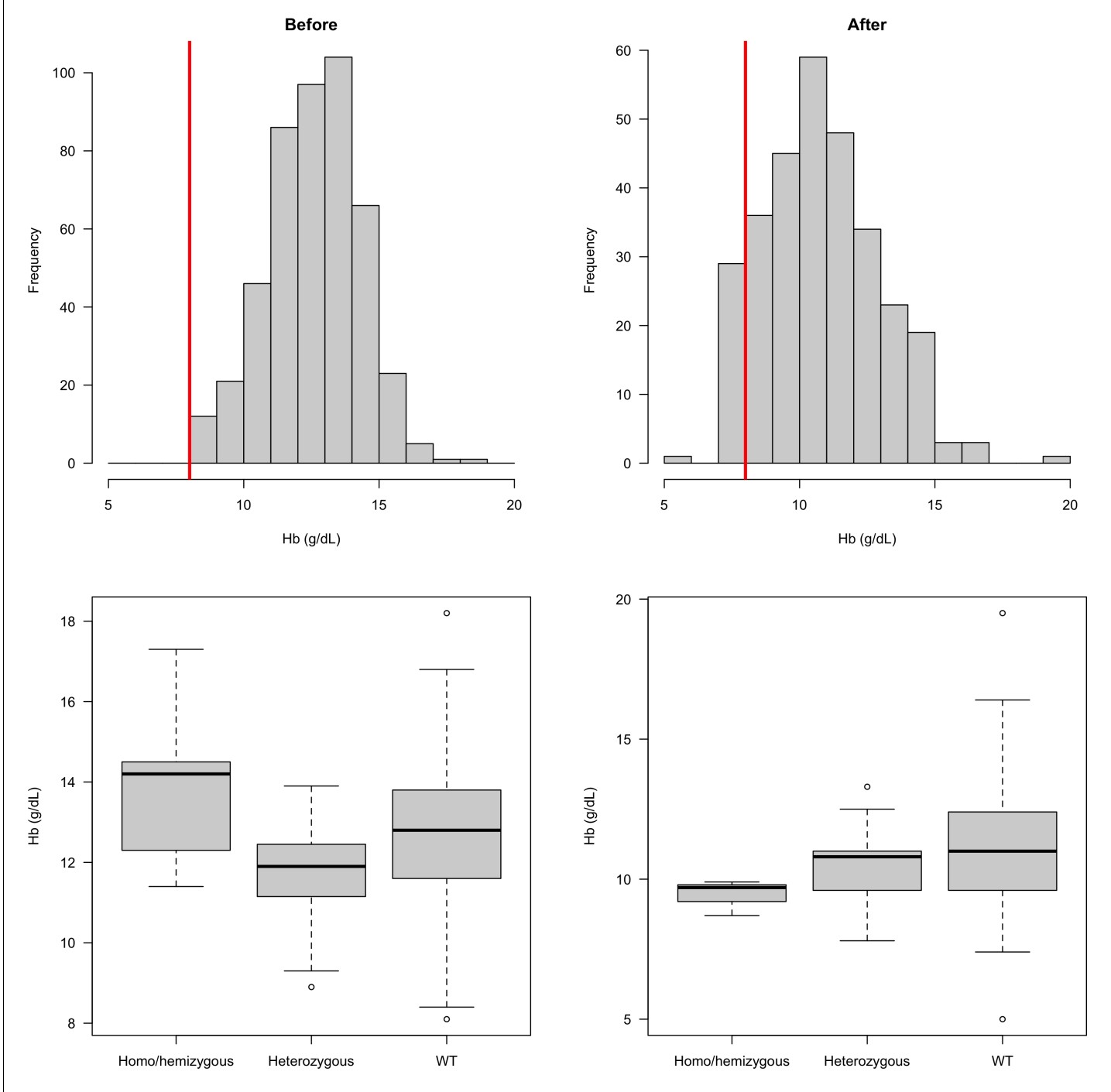

**Figure 3.** Distribution of admission haemoglobin concentrations in *P. vivax* malaria cases (overall: top row; stratified by G6PD genotype: bottom row). The top row shows the distribution of haemoglobin concentrations in the first part of the study (left panel, when there was a cut-off at 8 g/dL for inclusion) and in the second part of the study (right panel: after this inclusion criteria was relaxed). The bottom row shows these two distributions stratified by the patients' G6PD genotypes.

# Materials and methods

## Study area and participants

In Afghanistan, malaria is confined to the northern plains, the Jalalabad basin and river valleys, which fringe the central mountains to the west and south (*Figure 1*). Malaria cases begin presenting in

May, peak in July and August, and then decline with few cases seen after November. These studies were conducted over 10 years between 2009 and 2019.

The initial epidemiological study to assess the prevalence and genotypes of G6PD deficiency in Afghanistan was undertaken in one urban centre in each of nine provinces: Jalalabad (Nangarhar province), Mehtherlam (Laghman), Asadabad (Kunar), Maimana (Faryab), Taloqan (Takhar), Imamsahib (Kunduz), Pulikhumri (Baghlan), Pulialam (Lowgar), and the capital, Kabul (Kabul province) (*Jamornthanyawat et al., 2014*). Provinces were chosen on the basis of practical accessibility combined with having a high risk of vivax malaria (>1 case/1000 population/year), the exceptions being Kabul and Lowgar which have low malaria transmission. This analysis is restricted to the Pashtun ethnic group who live in the higher *P. vivax* transmission areas for two main reasons: the Pashtun comprises the majority of people in the Eastern provinces where malaria incidence is highest, and G6PD variants other than G6PD Med are rare in this ethnic group (*Bouma et al., 1995*; *Leslie et al., 2010*).

In the initial prevalence study in healthy subjects conducted in 2009, only males were studied (*Jamornthanyawat et al., 2014*). A convenience approach to sampling was used for reasons related to security. EDTA blood was obtained from healthy male adults and children attending outpatient medical laboratories with non-febrile illnesses or neighbouring vaccine administration who did not have malaria. Samples were transferred to filter-paper blood spots and stored in plastic zip-lock bags with silica gel at room temperature before transport to Bangkok for genotyping (*Jamornthanyawat et al., 2014*). In later prospective studies conducted in 2018 and 2019, subjects (both male and female) attending outpatients and those attending immunisation centres were sampled in the same way from the same sites in Nangarhar province as the clinical malaria studies (*Figure 1*).

## Clinical studies

Patients with symptomatic malaria presented to provincial malaria control centres adjacent to the Pakistan border. Jalalabad, the capital city of Nangarhar province, is a referral centre for the whole eastern region. Population movement takes place in both directions across this largely mountainous area bordering Pakistan. The malaria treatment protocols have been reported previously (*Awab et al., 2017*). Patients (aged >6 months) or older presenting with uncomplicated microscopy confirmed symptomatic vivax malaria were enrolled if they, or their carer, gave fully informed written informed consent. Inclusion criteria were monoinfection with *P. vivax*, axillary temperature $\geq$ 37.5°C or oral/rectal temperature $\geq$ 38°C, or history of fever in preceding 24 hr, and able to swallow oral medication and comply with study requirements. Exclusion criteria were any clinical or laboratory feature of severe malaria, significant comorbidity, known hypersensitivity to any study drugs, mixed species Plasmodium infection, and pregnancy or lactation. In the initial studies, patients with haemoglobin concentrations $\leq$ 8 g/dl were excluded, but in the later observational studies in 2019, anaemic patients were not excluded.

## Laboratory procedures

Capillary blood was collected for haemoglobin measurement (HemoCue), Giemsa staining of thick and thin blood films for parasite speciation and counts and G6PD testing. Before 2018 G6PD deficiency screening used the G-6-PD OSMMR 2000 (R and D Diagnostics) and NADPH fluorescent spot test kit (10 min incubation). From 2018, the CareStart rapid G6PD test was substituted. Dried blood spots were stored for later genotyping (*Leslie et al., 2010*; *Awab et al., 2017*). Genomic DNA was extracted using the QIAamp DNA Mini Kit (QIAGEN, Germany). Eluted genomic DNA samples were frozen at −20°C until PCR amplification. All samples were subjected to PCR-RFLP to assess the G6PD Mediterranean variant in exon 6, based on the protocol of *Samilchuk et al., 1999* with modifications including use of primers F14948 and R15158 at 250 nM, a 2 µl volume of each genomic DNA template, and a final reaction volume of 30 µl containing 20 mM Tris–HCl (pH 8.4), 50 mM KCl, 1 mM MgCl$_2$, 125 µM 4-deoxynucleotide triphosphate, and 0.05 units Platinum*Taq* DNA polymerase (Invitrogen, Brazil). After pre-denaturation at 95°C for 5 min, 45 PCR cycles were performed involving denaturation at 94°C for 1 min, annealing at 55°C for 1 min, and extension at 72°C for 1 min, with post-extension at 72°C for 7 min. Ten microlitres of each PCR product was digested with 10 units of the restriction enzyme *MboII* (New England Biolabs Inc) at 37°C for 3 hr and visualised by 2%

agarose gel electrophoresis. A product pattern of 104+98+28 bp indicates the G6PD Med mutation, whereas 202+28 bp indicates wild type. All PCR products of samples with mutant type and 10% of the wild-type samples were purified subsequently and the DNA was sequenced (Macrogen, Korea). Gene sequences of the PCR amplification products of *g6pd* (accession number: NM_001042351.2) were confirmed using the NCBI's Blastn and Blastx programmes. Gene polymorphisms were assessed by comparison with the reference sequences using BioEdit v7.2.5.

## Additional data included in the meta-analysis assessing the protective effect of G6PD deficiency

A systematic literature review identified two other sources of relevant data in Pashtun subjects. *Bouma et al., 1995* described a phenotyping survey in male Afghan and Pakistani Pashtun refugee schoolchildren (Safi and Mamund tribes, respectively). *Leslie et al., 2010* described a *P. vivax* case–control study in which patients who had been enrolled in earlier clinical trials assessing vivax malaria treatments were matched to community controls. This was also an Afghan refugee population. The refugee villages were located close to Peshawar in the North-West frontier province of Pakistan (*Figure 1*), close to the border with Afghanistan. The majority of individuals were also from the Safi and Mamund tribes. This study relied mainly on phenotyping (except for heterozygous females). In the meta-analysis, the phenotyping of males was assumed to be 100% sensitive and specific for the G6PD Med genotype. Thus, for the study reported by Leslie et al., we included all data from males, but for females, we only included data from genotyped individuals (154 genotyped female controls and 78 genotyped female cases) as phenotyping identifies only a minority of heterozygotes. The inclusion and exclusion criteria for the vivax cases reported by Leslie et al. were very similar to those in our clinical studies in Eastern Afghanistan.

## Statistical analysis

To estimate the degree of protection against vivax malaria and investigate a possible gene-dose effect, we analyzed all the available data under a Bayesian statistical model that estimates the relative risk reduction. We chose this model structure instead of a logistic regression with the allele as the predictive variable and the case–control status as the outcome (the usual model choice for genetic case–control studies) because we can assume that the protective effect in G6PD-deficient hemizygous males and homozygous females is the same (i.e. from a red cell perspective, they are identical). Log-odds estimated from a logistic regression model are not collapsible; therefore, this assumption cannot be input into a logistic regression model. Our formulation, in terms of relative reduction, allows for this assumption to be hard-wired into the model. It should be noted that this assumption applies to the marginal effect of G6PD deficiency, that is females could be more or less likely to have clinical vivax malaria relative to males, but the protective effect of hemizygote and homozygote G6PD deficiency is assumed identical, conditional on having clinical vivax malaria.

The model assumed that Hardy–Weinberg equilibrium held in the studied population(s). We tested this assumption by applying an exact test appropriate for X-linked biallelic markers in the control populations (p=0.9 for the data reported here; p=0.6 for the combined data in the meta-analysis, *Graffelman and Weir, 2016*). The unknown overall allele frequency of the Mediterranean variant of G6PD deficiency is denoted $p$. The total number of healthy males (controls) sampled is denoted $N_{males}^{healthy}$; the total number of *P. vivax* infected males (cases) is denoted $N_{males}^{vivax}$; within these groups, the number of hemizygote deficient males are denoted $N_{males}^{healthy,HemiD}$ and $N_{males}^{vivax,HemiD}$, respectively. The same notation is used for the female subjects, except that the superscript *HetD* refers to heterozygous deficient, *HomoD* refers to homozygous deficient, and *WT* refers to G6PD normal.

The likelihood of the observed data for the non-malaria controls is given by:

$$N_{males}^{healthy,HemiD} \sim Binomial\left(N_{males}^{healthy}, p\right)$$

$$\left\{N_{females}^{healthy,HomoD}, N_{females}^{healthy,HetD}, N_{females}^{healthy,WT}\right\} \sim Multinomial(\theta_{controls})$$

Where $\theta_{controls} = \left\{p^2, 2p[1-p], 1-p^2-2p[1-p]\right\}$.

The likelihood of the observed data for the vivax malaria cases is given by:

$$N_{males}^{vivax,\,HemiD} \sim Binomial\left(N_{males}^{vivax}, \alpha p\right)$$

$$\left\{N_{females}^{vivax,\,HomoD}, N_{females}^{vivax,\,HetD}, N_{females}^{vivax,\,WT}\right\} \sim Multinomial(\theta_{cases})$$

where $\theta_{cases} = \left\{\alpha p^2, \beta 2p[1-p], 1-\alpha p^2 - \beta 2p[1-p]\right\}$.

The parameter α, such that $0 \leq \alpha \leq 1$, denotes the vivax malaria protective effect of hemi- and homozygous deficiencies. A value of α = 1 implies no protective effect, and α = 0 implies a complete protective effect. Thus 1-α is the relative reduction in prevalence in hemi/homozygous G6PD-deficient *P. vivax* malaria cases compared to the controls. The parameter β, such that $0 \leq \beta \leq 1$, denotes the protective effect of heterozygous G6PD deficiency with the same interpretation.

Weakly informative Bayesian priors were subjectively chosen as follows:

$$\alpha \sim \text{Uniform}[0,1]$$

$$\beta \sim \text{Uniform}[0,1]$$

$$p \sim \text{Beta}(2,18)$$

This model allows for an additive effect (i.e. the protective effect in heterozygous females is less than in homo/hemizygotes) or an increased effect in heterozygotes (expected under a model of balancing selection driven by heterozygous advantage). The Bayesian model was written in Stan (**Stan Development Team, 2018**), and a reproducible implementation of the results in *rstan* is provided in the supplemental materials. Four independent chains were run for $10^6$ iterations, with the first half discarded for burn-in and second half thinned every 200 iterations, to give a total of 10,000 samples for the posterior distribution. Parameter estimates and uncertainty levels are reported as median posterior estimates with 95% CI (2.5% and 97.5% quantiles of the posterior distribution). Convergence of chains was checked visually with trace plots.

## Ethical approval

The clinical studies were approved by the Institutional Review Board of the Afghan Public Health Institute, Ministry of Public Health, Afghanistan, the Ethics Committee of the Faculty of Tropical Medicine, Mahidol University, Thailand, and the Oxford Tropical Research Ethics Committee, Oxford University, UK. The clinical trial was registered with the clinical trials website http://www.clinicaltrials.gov/ct under the identifier NCT01178021.

## Acknowledgements

The investigators are grateful to the doctors, laboratory technicians, and malaria workers in the sites in Afghanistan, the Afghanistan National Malaria and Leishmania Control Program (NMLCP), the two provincial health directorates, and the senior management of the Ministry of Public Health and WHO-Afghanistan. We are also very grateful to the Molecular Tropical Medicine and Genetics laboratory staff in the Faculty of Tropical Medicine, Mahidol University for their help with genotyping.

## Additional information

### Funding

| Funder | Grant reference number | Author |
| --- | --- | --- |
| Wellcome Trust | Principle Fellowship of NJ White 093956/Z/10/Z | Ghulam R Awab |
| Wellcome Trust | Major Overseas Programme-Thailand Unit Core Grant 093956/Z/10/Z | Kanokon Suwannasin |
| Wellcome Trust | Training Fellowship 107548Z/15/Z | Ghulam R Awab |

The funders had no role in study design, data collection and interpretation, or the decision to submit the work for publication.

## Author contributions
Ghulam R Awab, Resources, Data curation, Investigation, Project administration, Writing - review and editing; Fahima Aaram, Natsuda Jamornthanyawat, Kanokon Suwannasin, Data curation, Investigation; Watcharee Pagornrat, Data curation, Investigation, Methodology; James A Watson, Software, Formal analysis, Visualization, Writing - review and editing; Charles J Woodrow, Supervision, Investigation, Methodology, Project administration, Writing - review and editing; Arjen M Dondorp, Supervision, Funding acquisition, Writing - review and editing; Nicholas PJ Day, Supervision, Funding acquisition, Validation, Writing - review and editing; Mallika Imwong, Supervision, Investigation, Methodology; Nicholas J White, Conceptualization, Resources, Supervision, Investigation, Methodology, Writing - original draft, Project administration, Writing - review and editing

## Author ORCIDs
James A Watson (iD) https://orcid.org/0000-0001-5524-0325
Arjen M Dondorp (iD) http://orcid.org/0000-0001-5190-2395
Nicholas PJ Day (iD) https://orcid.org/0000-0003-2309-1171
Nicholas J White (iD) https://orcid.org/0000-0002-1897-1978

## Ethics
Clinical trial registration Clinicaltrials.gov NCT01178021.
Human subjects: The clinical studies were approved by the Institutional Review Board of the Afghan Public Health Institute, Ministry of Public Health, Afghanistan, the Ethics Committee of the Faculty of Tropical Medicine, Mahidol University, Thailand, and the Oxford Tropical Research Ethics Committee, Oxford University, UK.

## Decision letter and Author response
Decision letter https://doi.org/10.7554/eLife.62448.sa1
Author response https://doi.org/10.7554/eLife.62448.sa2

# Additional files
## Supplementary files
- Source code 1. Contains source code and data.
- Transparent reporting form

## Data availability
All data used in the analysis are available along with the code which is given in the supplementary materials.

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
