## [Decision Letter]

**Acceptance summary:**

The exact relationship between G6PD deficiency and malaria protection remains uncertain. This study provides evidence that the G6PD Med mutation (563 C>T) protects against clinical *Plasmodium vivax* disease. It uses a Bayesian statistical approach which specifically elucidates the particular protection which female heterozygotes versus male hemizygotes (or female homozygotes) for the Med mutation may experience. This is an important contribution to our understanding of the relationship between G6PD deficiency and *P. vivax*.

**Decision letter after peer review:**

Thank you for submitting your article "Protective effect of Mediterranean type glucose-6-phosphate dehydrogenase deficiency against *Plasmodium vivax* malaria" for consideration by *eLife*. Your article has been reviewed by three peer reviewers, and the evaluation has been overseen by a Reviewing Editor and a Senior Editor. The following individuals involved in review of your submission have agreed to reveal their identity: Bridget S Penman (Reviewer #1); Anirban Mondal (Reviewer #2).

The reviewers have discussed the reviews with one another and the Reviewing Editor has drafted this decision to help you prepare a revised submission.

Summary

The exact relationship between G6PD deficiency and malaria protection remains uncertain. This study provides evidence that the G6PD Med mutation (563 C>T) protects against clinical *Plasmodium vivax* disease. It uses a Bayesian statistical approach which specifically elucidates the particular protection which female heterozygotes versus male hemizygotes (or female homozygotes) for the Med mutation may experience. This is an important contribution to our understanding of the relationship between G6PD deficiency and *P. vivax*.

Overall, the reviewers were positive about the work and its potential, but have some clear concerns that will require additional data, analyses, and interpretation. Below are the main points raised by the reviewers that would need to be addressed to for a revised manuscript.

1) The presence of mixed infections: although the work is focused on *P. vivax*, the majority (95%) of malaria in Afghanistan is caused by *P. falciparum* that means mixed species infections are likely high and *P. falciparum* infections may be obscuring *P. vivax* infections. It is not clear to what extent G6PD deficiency may impact the chance of being coinfected with both falciparum and vivax. Ideally, PCR verification of these samples would be performed to confirm the species for samples included in the analysis. Without this molecular data, the overall assessments of susceptibility to vivax malaria in association with G6PD Med is incomplete.

2) The analysis relies on a number of assumptions made about Pashtun population genetics (e.g. is it reasonable to assume the same frequency of the relevant mutation throughout all the tribes in the study, and should this be at Hardy Weinberg equilibrium?) and it is not clear to what extent these assumptions are justified since little evidence/support is provided. In particular, the assumptions about Hardy Weinberg equilibrium of G6PD Med within the Pashtun population need to be justified and supported since the analysis is highly reliant on this assumption.

3) The exclusion criteria does not appear to have been uniformly applied – in particular anemia was an exclusion criteria for only part of the data. This was not clear and may impact the overall significance of statistical results.

4) While the manuscript makes a number of conclusions about female homozygotes, these are not strongly supported by the evidence. In particular, the study is likely under-powered with regard to clinical associations among female homozygotes with G6PD Med, but this is not addressed and the stated conclusions are likely stronger than what can be supported by the data/analyses provided.

Reviewer #1:

There are aspects of the study which I would like to see explained further before I can be certain of the strength of the evidence.

1) The key result rests on the fact that the proportion of G6PD deficient individuals is lower among patients with vivax malaria than in controls without malaria. However, as the authors note: "In the initial clinical studies reported here severe anaemia was an exclusion criterion and this could have reduced the proportion of G6PD deficient patients [among malaria patients – my insertion]". To enable readers to judge for themselves how big an effect this could have had, it would be useful to indicate the numbers of patients affected -e.g. by providing a supplementary table which splits up the data in table 1 according to before and after the anaemia criterion was changed. It may also be useful to conduct a supplementary analysis among just the data where severe anaemia was not an exclusion criterion – but I do not want to request that without knowing the numbers involved.

2) The Leslie study [14], from which data was included in the meta-analysis had "372 cases and 743 controls", but from the numbers in Table 1 it seems that not all of these have been included in the present meta-analysis. The Materials and methods need to make clear which of the samples from the Leslie study were included in the meta-analysis, and why. I can imagine this may be something to do with the geographical origins of the refugee populations in the Leslie study, since this study focused on high malaria transmission regions only, but this needs to be explained.

3) The method used here implicitly assumes that there is little heterogeneity in G6PD Med frequencies between different Pashtun tribal groups currently living in malaria endemic regions. It is hard to know whether or not this is a reasonable assumption, because inevitably the evidence for G6PD Med frequencies in different tribal groups is based on small numbers (e.g. the Leslie study [14]). The authors do acknowledge this with the statement in the Discussion that "genetic heterogeneity within the Pashtun group" may be a confounder. However, I would like to see this potential limitation of the study discussed in more detail in a specific section of the Discussion or Introduction, including reasons why it is reasonable to assume a consistent Med frequency. Are there any other sources of evidence about genetic diversity within the Pashtun group, e.g. from studies at other loci?

4) The Materials and methods explain the case definition for *P. vivax* infection for the majority of the data (that which was obtained by the authors). However, it is not stated that the case definition for *P. vivax* infection was equivalent in the Leslie study, from which data was included in the meta-analysis.

5) The number of female homozygotes in this study is (inevitably) very small. I do not object to the authors including female homozygotes and male hemizygotes in the same category, I can see the biological logic behind it, and I like their entire approach to estimate the gene dose effect. However, on the other hand, surely it remains possible that, despite their similar red blood cells, a female homozygote for G6PD Med may not be exactly the same as a male hemizygote for G6PD Med with respect to their likelihood of displaying clinical vivax malaria. If the authors had fitted α, β and γ (for 3 different genotypes, with γ referring to the female homozygote), it seems likely to me that α and β would have been similar to the current estimates and γ would have had a huge credible interval because there is little data. If the authors concur, it would be worth acknowledging this explicitly in the Discussion – because to me this means that this study tells us quite a lot about male hemizygotes and female heterozygotes, but doesn't tell us much about female homozygotes.

Reviewer #2:

This review is mainly focussed on the statistical analysis part of the article.

The authors have used a well-established and standard Bayesian methodology for inference on the role of G6PD Med as a protective effect against vivax malaria. The Bayesian model is appropriate assuming that the Hardy-Weinberg equilibrium is held in the studied population(s). A Markov Chain Monte Carlo sampling based inference is used for the Bayesian inference.

I have the following concerns and/or comments which should be addressed.

1) The author states that weakly informative Bayesian priors were used in the model. But this is not true for the allele frequency parameter p. I believe Β(2,18) is a strongly informative prior for p. Some explanation for choosing such a strong prior for p should be included in the article. How are the values for hyperparameters (2 and 18) chosen? How sensitive are the results with respect to these hyperparameter values?

2) It is desired that the authors include some plots and statistics assessing the convergence of the MCMC method, e.g. trace plots of the posterior samples, the Gelman-Rubin statistics (R_hat_), in the R-markdown document.

3) The Bayesian model assumed that the Hardy-Weinberg equilibrium is held in the studied population(s). Can the authors discuss why this assumption is reasonable in the studied population(s) and how the model can be modified if the equilibrium is disturbed by mutations, natural selection and non-random mating?

Reviewer #3:

The Short Report article suggests that the Mediterranean variant of G6PD deficiency confers a strong gene-dose proportional protective effect against symptomatic *Plasmodium vivax* malaria in the Pashtun ethnic group in Afghanistan. Results suggesting that G6PDd Med hemizygous males and homozygous females have 67% relative reduction against acute *P. vivax* malaria and G6PDd Med heterozygous females have 51% relative reduction is of particular interest for malaria and human genetics researchers. The authors present a very thoughtful discussion of *P. vivax* sensitivity to G6PD deficiency oxidant effects on parasite viability.

1) The authors assume that Hardy-Weinberg equilibrium exists in the populations studied but they provided no evidence that it exists. Please provide the evidence that Hardy-Weinberg equilibrium exists for the alleles/genotypes for G6PD. Please also discuss how a deviation from Hardy-Weinberg equilibrium would affect the results of this study.

2) Previous studies have shown that the G6PD Mediterranean allele is the predominant G6PD deficiency allele in the region. The G6PD gene is, however quite polymorphic, with over 180 mutations and at least two different G6PD Mediterranean allelic haplotypes. Thus the possibility that further genetic polymorphism is associated with G6PD deficiency and significant potential for population mixing (particularly in large cosmopolitan urban settings) may introduce genetic diversity in the G6PD gene. Please discuss how additional G6PD deficiency alleles could affect the results of this study.

3) Brief details were provided on malaria diagnosis by Giemsa-stained thick and thin blood films. *P. falciparum* accounts for 95% of malaria cases in Afghanistan; *P. vivax* 5% (http://www.emro.who.int/afg/programmes/malaria-leishmaniasis.html). Diagnosis of *P. vivax* and accurate assessment of mixed species infections in areas where these two species are co-endemic is known to confound malaria microscopy. Additionally, submicroscopic Plasmodium species infections (involving *P. vivax* in particular), revealed by parallel PCR methods, is now very well established. Please provide further details as to how Plasmodium species diagnosis was performed, how monoinfection by *P. vivax* was confirmed and how exclusion of other Plasmodium species and/or mixed species infections were ruled out. If the complexity of Plasmodium species infections has been confirmed by PCR, the authors should revise their Laboratory Procedures section and present their data in these Result section. If Plasmodium species infections have not been confirmed by PCR, the authors should perform this work.

[Editors' note: further revisions were suggested prior to acceptance, as described below.]

Thank you for resubmitting your work entitled "Protective effect of Mediterranean type glucose-6-phosphate dehydrogenase deficiency against *Plasmodium vivax* malaria" for further consideration by *eLife*. Your revised article has been evaluated by a Senior Editor and a Reviewing Editor.

The manuscript has been improved but there are some remaining issues that need to be addressed before acceptance, as outlined below:

1) Include information in the main text about co-infection. The Who data seems to suggest that co-infection may be possible, although unlikely, and this is not likely to impact the study's outcome. However, the authors should address this point with a few sentences in the manuscript since readers may be concerned by the absence of PCR confirmation.

2) Include a figure to highlight/provide more context/information about the data on the haemoglobin levels in the different genotypes. While the code provide is helpful, it would be much easier for the reader to interpret if it was summarized with a figure/legend. Also, all figures/etc. should be labeled and well references in the supplement.

3) We propose the following smaller edits (suggestions in capitals):

In the Abstract: "Thus, in THE absence of very large population stratification within the Pashtun ethnic group (i.e. confounding these results), the G6PD Med35 genotype confers a very large and gene dose proportional protective effect against acute vivax malaria. "

In the Discussion, when the protective effects of G6PD deficiency against falciparum are discussed, this sentence "It is possible that G6PD deficiency confers no significant protection against infection by *P. falciparum*, but protects only against life threatening illness." could be improved by replacing "infection by *P. falciparum*" with something like "mild falciparum disease" or "uncomplicated clinical falciparum malaria".… just because protecting against "infection" could also mean protecting against completely asymptomatic infection, whereas the bulk of the evidence that has been discussed relates to disease with clinical symptoms.

---

## [Author Response]

Overall, the reviewers were positive about the work and its potential, but have some clear concerns that will require additional data, analyses, and interpretation. Below are the main points raised by the reviewers that would need to be addressed to for a revised manuscript.1) The presence of mixed infections: although the work is focused on P. vivax, the majority (95%) of malaria in Afghanistan is caused by *P. falciparum* that means mixed species infections are likely high and *P. falciparum* infections may be obscuring P. vivax infections. It is not clear to what extent G6PD deficiency may impact the chance of being coinfected with both falciparum and vivax. Ideally, PCR verification of these samples would be performed to confirm the species for samples included in the analysis. Without this molecular data, the overall assessments of susceptibility to vivax malaria in association with G6PD Med is incomplete.

We understand this reasonable concern but this description of the epidemiology in Afghanistan is wrong. It is based upon incorrect information provided (unfortunately) on the WHO EMRO site. The author of the WHO situation report may have got *P. falciparum* and *P. vivax* mixed up? The World Malaria report (2020) gives tells the true story. In fact, *P. falciparum* has now almost gone from Afghanistan, and was rare during the course of this study. It is certainly not a confounder.

2) The analysis relies on a number of assumptions made about Pashtun population genetics (e.g. is it reasonable to assume the same frequency of the relevant mutation throughout all the tribes in the study, and should this be at Hardy Weinberg equilibrium?) and it is not clear to what extent these assumptions are justified since little evidence/support is provided. In particular, the assumptions about Hardy Weinberg equilibrium of G6PD Med within the Pashtun population need to be justified and supported since the analysis is highly reliant on this assumption.

We agree that this is an important point and population stratification between cases and controls could bias the effect estimates. We cannot adjust for population stratification as we do not have other genetic marker data. We have added to the section in the Discussion outlining this limitation. However, the population stratification would have to be very large to account for such a large protective effect. The data from Leslie et al. (from a geographic location close to the current study sites, albeit with a refugee population – see map shown in Figure 1) do show that there are different allele frequencies between the different tribes comprising the Pasthtun ethnic group. We do not have data on case-control allele counts broken down by tribe, so we cannot adjust for this. However, we note that this does not make the model mis-specified per se (as the likelihood is still binomial for the aggregate data). The data from Leslie et al. are carefully matched so are no large differences between cases and controls. It is reassuring that the effect sizes between our data and the data from Leslie are very similar.

We have added a statistical test for deviations from Hardy –Weiberg equilibrium (H-W). This shows no evidence for a departure from H-W in the controls (i.e. data from the controls are entirely consistent with the H-W assumption). This is referred to in the Materials and methods and given in the Source code 1 file.

3) The exclusion criteria does not appear to have been uniformly applied – in particular anemia was an exclusion criteria for only part of the data. This was not clear and may impact the overall significance of statistical results.

We fully accept this criticism of our retrospective study, which we did acknowledge in the paper. We now present a sub-analysis to see if there are significant differences between the two groups (i.e. before versus after: when the exclusion criterion was applied versus afterwards when it was lifted) which might have affected our overall estimates. This shows that the 8g/dL criterion excludes about 10% of patients (and would have excluded 1 of the 10 G6PD deficient patients – a female heterozygote) if applied to the later series. There is no evidence that the exclusion criteria would have affected the effect estimates as there was no difference in the distribution of haemoglobin concentrations between G6PD deficient and G6PD normal patients (analysis now included).

4) While the manuscript makes a number of conclusions about female homozygotes, these are not strongly supported by the evidence. In particular, the study is likely under-powered with regard to clinical associations among female homozygotes with G6PD Med, but this is not addressed and the stated conclusions are likely stronger than what can be supported by the data/analyses provided.

This is correct-the number of female homozygotes is very small, as expected. We are unaware of any evidence that the red cells of hemizygotes and homozygotes are phenotypically different. The assumption that red cells of female homozygotes and male hemizygotes are identical is fully supported by the underlying biology (due to Lyonisation). The model only makes an assumption about relative effects (hemi or homo-zygous deficiency versus WT), so males and females could differ in their absolute risk but this would not affect the model estimates unless there was some other genetic variation that had a direct interaction with G6PD deficiency. Pooling of male hemizygotes and female homozygotes is now the norm for the G6PD deficiency (otherwise we could never say anything about female homozygotes, see for example the recent paper from Kenya: https://ashpublications.org/bloodadvances/article/4/23/5942/474381/Glucose-6-phosphate-dehydrogenase-deficiency-and ). It also the norm for general tests of association on the X-chromosome, see https://pubmed.ncbi.nlm.nih.gov/18441336/. Given the strong background biological knowledge, the rationale for not pooling hemizygotes and homozygotes is unclear.

Reviewer #1:There are aspects of the study which I would like to see explained further before I can be certain of the strength of the evidence.1) The key result rests on the fact that the proportion of G6PD deficient individuals is lower among patients with vivax malaria than in controls without malaria. However, as the authors note: "In the initial clinical studies reported here severe anaemia was an exclusion criterion and this could have reduced the proportion of G6PD deficient patients [among malaria patients – my insertion]". To enable readers to judge for themselves how big an effect this could have had, it would be useful to indicate the numbers of patients affected -e.g. by providing a supplementary table which splits up the data in Table 1 according to before and after the anaemia criterion was changed. It may also be useful to conduct a supplementary analysis among just the data where severe anaemia was not an exclusion criterion – but I do not want to request that without knowing the numbers involved.

We present a supplementary analysis split as requested.

2) The Leslie study [14], from which data was included in the meta-analysis had "372 cases and 743 controls", but from the numbers in Table 1 it seems that not all of these have been included in the present meta-analysis. The Materials and methods need to make clear which of the samples from the Leslie study were included in the meta-analysis, and why. I can imagine this may be something to do with the geographical origins of the refugee populations in the Leslie study, since this study focused on high malaria transmission regions only, but this needs to be explained.

This is partly a mistake on our part and partly a lack of clarity. We had only included data from Leslie et al. for the individuals who had been genotyped (418 samples). However, genotyping data are only necessary for females (where the phenotypic test cannot determine heterozygosity). Therefore, we can augment the total number of samples in the meta-analysis to 316 male controls and 157 male cases (total of 703 cases and controls). We can only use data from the genotyped females reported in Leslie for the meta-analysis (124 controls and 78 cases). We have added one line to the Materials and methods to clarify this.

3) The method used here implicitly assumes that there is little heterogeneity in G6PD Med frequencies between different Pashtun tribal groups currently living in malaria endemic regions. It is hard to know whether or not this is a reasonable assumption, because inevitably the evidence for G6PD Med frequencies in different tribal groups is based on small numbers (e.g. the Leslie study [14]). The authors do acknowledge this with the statement in the Discussion that "genetic heterogeneity within the Pashtun group" may be a confounder. However, I would like to see this potential limitation of the study discussed in more detail in a specific section of the Discussion or Introduction, including reasons why it is reasonable to assume a consistent Med frequency. Are there any other sources of evidence about genetic diversity within the Pashtun group, e.g. from studies at other loci?

We accept that this is a limitation. As pointed out above the populations studied in this investigation, and those reported earlier by Leslie et al. were located in a similar geographic area divided by the Afghanistan-Pakistan border. There are few population genetic studies of the Pashtun ethnic group, but there is evidence that it is a diverse group (https://journals.plos.org/plosone/article?id=10.1371/journal.pone.0076748). Having additional genetic marker data would allow for adjustment for population stratification but this is not available.

4) The Materials and methods explain the case definition for P. vivax infection for the majority of the data (that which was obtained by the authors). However, it is not stated that the case definition for P. vivax infection was equivalent in the Leslie study, from which data was included in the meta-analysis.

Inclusion criteria in the Leslie et al. clinical trial (which contributed cases to their retrospective case-control evaluation) were microscopically confirmed *P vivax* malaria, age 3 years or older, written or witnessed verbal consent (by parents in the case of minors), available for the duration of follow-up, and willingness to be tested for G6PD deficiency at admission. Exclusion criteria were general condition requiring hospital admission, evidence of any concomitant infection or disease likely to mask treatment response, known allergy to any classes of the study drugs, known methemoglobin reductase deficiency, treatment within the past 7 days with any drug with known antimalarial properties or with an investigational drug within 30 days, severe anaemia (haemoglobin <7 g/dL), mixed infection (*P vivax* and *P falciparum*), and pregnancy (confirmed by testing), lactation, or both. Thus, the criteria, case definition and the patient populations were all very similar to the current studies by Awab et al. with which they were pooled.

5) The number of female homozygotes in this study is (inevitably) very small. I do not object to the authors including female homozygotes and male hemizygotes in the same category, I can see the biological logic behind it, and I like their entire approach to estimate the gene dose effect. However, on the other hand, surely it remains possible that, despite their similar red blood cells, a female homozygote for G6PD Med may not be exactly the same as a male hemizygote for G6PD Med with respect to their likelihood of displaying clinical vivax malaria. If the authors had fitted α, β and γ (for 3 different genotypes, with γ referring to the female homozygote), it seems likely to me that α and β would have been similar to the current estimates and γ would have had a huge credible interval because there is little data. If the authors concur, it would be worth acknowledging this explicitly in the Discussion – because to me this means that this study tells us quite a lot about male hemizygotes and female heterozygotes, but doesn't tell us much about female homozygotes.

We agree that the study conclusions regarding the fully deficient state are almost entirely based on males hemizygotes. However, the model assumption is about a marginal relative effect: it could be the case that female homozygotes and male hemizygotes do not have the same absolute probability of having clinical vivax (for reasons unrelated to G6PD deficiency). However, the model assumption is that, conditional on having clinical malaria, the protective effect of the hemizygote and homozygotes states are equal. There is no evidence that this would not be the case but, as mentioned, the data are too sparse to support this claim (the biological prior takes precedence over the sparse information in the data). We have one sentence in Discussion making this assumption explicit.

Reviewer #2:This review is mainly focussed on the statistical analysis part of the article.The authors have used a well-established and standard Bayesian methodology for inference on the role of G6PD Med as a protective effect against vivax malaria. The Bayesian model is appropriate assuming that the Hardy-Weinberg equilibrium is held in the studied population(s). A Markov Chain Monte Carlo sampling based inference is used for the Bayesian inference.I have the following concerns and/or comments which should be addressed.1) The author states that weakly informative Bayesian priors were used in the model. But this is not true for the allele frequency parameter p. I believe Β(2,18) is a strongly informative prior for p. Some explanation for choosing such a strong prior for p should be included in the article. How are the values for hyperparameters (2 and 18) chosen? How sensitive are the results with respect to these hyperparameter values?

We disagree that this is a strong prior. The Β(2,18) distribution has 80% of its mass between 2.8% and 19% (median value is 8.6%) Given our prior knowledge (i.e. G6PDd Med in Afghanistan has around 5-10% allele frequency), this fits the standard usage of the “weakly informative”. A plot is shown for illustration in Author response image 1.

**Author response image 1. sa2fig1:** 

2) It is desired that the authors include some plots and statistics assessing the convergence of the MCMC method, e.g. trace plots of the posterior samples, the Gelman-Rubin statistics (R_hat_), in the R-markdown document.

The model converges very well given its simplicity (unimodal distribution). Traceplots are now included in the RMarkdown and referred to in the Materials and methods.

3) The Bayesian model assumed that the Hardy-Weinberg equilibrium is held in the studied population(s). Can the authors discuss why this assumption is reasonable in the studied population(s) and how the model can be modified if the equilibrium is disturbed by mutations, natural selection and non-random mating?

We now include standard tests for H-W equilibrium in the RMarkdown (reported in the main text as well). There is no evidence in the controls for departure from H-W.

Reviewer #3:The Short Report article suggests that the Mediterranean variant of G6PD deficiency confers a strong gene-dose proportional protective effect against symptomatic Plasmodium vivax malaria in the Pashtun ethnic group in Afghanistan. Results suggesting that G6PDd Med hemizygous males and homozygous females have 67% relative reduction against acute P. vivax malaria and G6PDd Med heterozygous females have 51% relative reduction is of particular interest for malaria and human genetics researchers. The authors present a very thoughtful discussion of P. vivax sensitivity to G6PD deficiency oxidant effects on parasite viability.1) The authors assume that Hardy-Weinberg equilibrium exists in the populations studied but they provided no evidence that it exists. Please provide the evidence that Hardy-Weinberg equilibrium exists for the alleles/genotypes for G6PD. Please also discuss how a deviation from Hardy-Weinberg equilibrium would affect the results of this study.

As above. There is no evidence for large departures from H-W.

2) Previous studies have shown that the G6PD Mediterranean allele is the predominant G6PD deficiency allele in the region. The G6PD gene is, however quite polymorphic, with over 180 mutations and at least two different G6PD Mediterranean allelic haplotypes. Thus the possibility that further genetic polymorphism is associated with G6PD deficiency and significant potential for population mixing (particularly in large cosmopolitan urban settings) may introduce genetic diversity in the G6PD gene. Please discuss how additional G6PD deficiency alleles could affect the results of this study.

Previous studies have only found the G6PD Mediterranean variant in the Pashtun, but these have not been exhaustive. If additional G6PD deficiency variants did occur in this study they would have diluted the protective effect estimate. We have added a sentence in the Discussion.

3) Brief details were provided on malaria diagnosis by Giemsa-stained thick and thin blood films. *P. falciparum* accounts for 95% of malaria cases in Afghanistan; P. vivax 5% (http://www.emro.who.int/afg/programmes/malaria-leishmaniasis.html). Diagnosis of P. vivax and accurate assessment of mixed species infections in areas where these two species are co-endemic is known to confound malaria microscopy. Additionally, submicroscopic Plasmodium species infections (involving P. vivax in particular), revealed by parallel PCR methods, is now very well established. Please provide further details as to how Plasmodium species diagnosis was performed, how monoinfection by P. vivax was confirmed and how exclusion of other Plasmodium species and/or mixed species infections were ruled out. If the complexity of Plasmodium species infections has been confirmed by PCR, the authors should revise their Laboratory Procedures section and present their data in these Result section. If Plasmodium species infections have not been confirmed by PCR, the authors should perform this work.

This is an unfortunate mistake by WHO EMRO.